# Costs and cost-effectiveness of a comprehensive tuberculosis case finding strategy in Zambia

Youngji Jo[1], Mary Kagujje[2], Karl Johnson[3], David Dowdy[1], Peter Hangoma[4], Lophina Chiliukutu[2], Monde Muyoyeta[2], Hojoon Sohn[1] *

1 Johns Hopkins Bloomberg School of Public Health, Baltimore, Maryland, United States of America, 2 Centre For Infectious Disease Research in Zambia (CIDRZ), Lusaka, Zambia, 3 University of North Carolina School of Global Public Health, Chapel Hill, North Carolina, United States of America, 4 School of Public Health, University of Zambia, Lusaka, Zambia

* hsohn6@jhu.edu

**Data Availability Statement:** All relevant data are within the manuscript and its Supporting information files.

## Abstract

### Introduction

Active-case finding (ACF) programs have an important role in addressing case detection gaps and halting tuberculosis (TB) transmission. Evidence is limited on the cost-effectiveness of ACF interventions, particularly on how their value is impacted by different operational, epidemiological and patient care-seeking patterns.

### Methods

We evaluated the costs and cost-effectiveness of a combined facility and community-based ACF intervention in Zambia that utilized mobile chest X-ray with computer-aided reading/ interpretation software and laboratory-based Xpert MTB/RIF testing. Programmatic costs (in 2018 US dollars) were assessed from the health system perspective using prospectively collected cost and operational data. Cost-effectiveness of the ACF intervention was assessed as the incremental cost per TB death averted over a five-year time horizon using a multi-stage Markov state-transition model reflecting patient symptom-associated care-seeking and TB care under ACF compared to passive care.

### Results

Over 18 months of field operations, the ACF intervention costed $435 to diagnose and initiate treatment for one person with TB. After accounting for patient symptom-associated care-seeking patterns in Zambia, we estimate that this one-time ACF intervention would incrementally diagnose 407 (7,207 versus 6,800) TB patients and avert 502 (611 versus 1,113) TB-associated deaths compared to the status quo (passive case finding), at an incremental cost of $2,284 per death averted over the next five-year period. HIV/TB mortality rate, patient symptom-associated care-seeking probabilities in the absence of ACF, and the costs of ACF patient screening were key drivers of cost-effectiveness.

**Funding:** This study was funded by the Stop TB Partnership at the UNOPS through the TB REACH wave 5 grant. TB REACH – an initiative of Stop TB Partnership – is funded by Global Affairs Canada [grant number CA-3-D000920001] and The Bill and Melinda Gates Foundation [OPP1139029]. This study was also funded by the Korea Health Technology Research and Development Project through support from the Korea Health Industry Development Institute and the Ministry of Health and Welfare, Republic of Korea, in the form of funds to HS [H19C1235]. The funders had no role in study design, data collection and analysis, decision to publish, or preparation of the manuscript.

**Competing interests:** The authors have declared that no competing interests exist.

**Abbreviations:** ACF, Active-case finding; CHW, community health workers; ICER, incremental cost-effectiveness ratio; PCF, Passive Case Finding; PLWH, people living with human immunodeficiency virus; TB, Tuberculosis; WTP, willingness-to-pay.

## Conclusions

A one-time comprehensive ACF intervention simultaneously operating in public health clinics and corresponding catchment communities can have important medium-term impact on case-finding and be cost-effective in Zambia. The value of such interventions increases if targeted to populations with high HIV/TB mortality, substantial barriers (both behavioral and physical) to care-seeking exist, and when ACF interventions can optimize screening by achieving operational efficiency.

## Introduction

Annually, an estimated 30% of tuberculosis (TB) incident cases are not notified, and this large pool of undetected diseases fuels ongoing transmission [1]. Missed diagnosis often results from underlying economic, geographic, socio-cultural and health system barriers to accessing TB care [2, 3]. Particularly, there remain large gaps in both the detection and treatment of HIV-associated TB; out of the almost 815,000 new cases of TB among people living with HIV (PLWH) in 2019, only 56% were diagnosed and reported [1]. As TB symptoms in this population may be nonspecific, many PLWH with TB may seek care late in their disease course [4]. Moreover, diagnosis of TB in PLWH is complicated by atypical chest radiography and a higher prevalence of sputum-smear negative disease [5–7]. In 2019, Zambia had an estimated 59,000 incident TB cases (333 per 100,000 per year), and of these, 28,000 were co-infected with HIV [1].

Given patient-level and health system barriers to accessing TB care [8, 9], improvements in routine TB diagnostic infrastructure alone (e.g. scaling-up rapid molecular tests) may not be sufficient to strengthen the TB care cascade in high TB burden settings such as Zambia. Active case finding (ACF) strategies (e.g. mass chest X-ray screening, household surveys, outpatient symptom screening, targeting high-risk populations) may therefore have an important role in facilitating early case detection and addressing gaps in the TB care cascade [10, 11]. To comprehensively address case detection gaps and improve linkage to care, the Centre for Infectious Disease Research in Zambia (CIDRZ) developed and implemented a one-time intensive active-case finding (ACF) program in both a health facility and the surrounding communities [12]. This combined ACF strategy included programmatic components to improve community awareness, access to timely diagnosis using mobile X-ray screening and rapid molecular testing (Xpert MTB/RIF), and facilitate linkage to care, including further clinical evaluation or initiation of TB treatment.

The resource-intensive nature of ACF interventions represents a major barrier to implementation and scale-up [13]. Previous modeling studies have shown that TB prevalence [14], human resource constraints [15], patients' care seeking behavior [16], and linkage to care [17] may substantially affect the cost-effectiveness of ACF. As such, costs and cost-effectiveness of ACF will depend on how intervention components are designed, integrated, and implemented. Moreover, it is also important to understand how a range of local epidemiological and operational factors such as TB prevalence, underlying patient care-seeking behaviors, health system capacity, and community acceptance can influence programs' scope, operations, and value [18].

In this study, we empirically assessed costs of key programmatic components of the CIDRZ ACF program. Then, we used a multi-stage Markov state-transition model representing

patients' symptom-associated care-seeking patterns and corresponding TB care algorithms to estimate cost-effectiveness if a similar intervention were to be implemented in a setting with TB epidemiology and economic conditions consistent with national averages.

## Methods

### Study setting and operations

The CIDRZ ACF project operated in the catchment area of the George primary health care centre (GPHC, covering a peri-urban settlement of 172,550 people), a Lusaka province public-sector facility offering TB diagnostic and treatment services. The intervention was operationalized by two distinct teams—community-based outreach and facility-based—which screened individuals for presumptive TB and facilitated linkage to care for diagnosis and treatment of TB. Diagnostic procedures included mobile X-ray followed by laboratory-based Xpert MTB/RIF (Xpert) testing. The community-based outreach team (which rotated through different areas of the GPHC catchment region, 3–4 days per area) employed a group of trained community health workers (CHWs) and drama teams who conducted community TB awareness events, community and door-to-door symptom screening, and chest X-rays taken in a mobile X-ray (mCXR) unit installed in a truck and interpreted by a computer-aided reading/interpretation software (CAD4TB Version 1.5, Delft Images, *hereafter* CAD4TB). The facility-based team operated out of an open-access tent at GPHC where four trained staff members conducted TB symptom screening and patient referral to TB services (including treatment initiation). In addition to screening people presenting (or referred) to the tent, the facility-based team also made regular visits to the antiretroviral therapy (ART) clinic and other departments (e.g., maternal health or general outpatient clinics) at GPHC to identify patients with TB symptoms and/or those patients otherwise indicated for TB screening (e.g., people living with HIV). Both community-based outreach and facility-based team collected sputum samples on-spot from the patients showing abnormal symptoms or Chest Xray results for Xpert testing in the GPHC laboratory once a patient was identified as a presumptive TB patient after initial symptom and mCXR screening. When on-site mCXR was not available (either in-use by one of the teams or inoperable), all symptomatic clients received Xpert without CXR screening. Patients with a positive Xpert test result were immediately followed-up by the ACF team and referred to the TB clinic for treatment initiation.

### Cost analysis

Cost and resource-use data were collected using a standardized cost data collection and analysis tool developed by our team. This tool allows for cost and resource-use data collection by key activity component (i.e. training, screening, diagnosis, and treatment) for analysis using a top-down method [19]. Human resource costs for each activity component were estimated based on their estimated level of effort (LOE), approximated as proportional time allocation (%) of their full-time work spent on each activity during the program operations, assessed periodically (each quarter) using a workload survey. For the costs of goods, equipment, and services, we divided direct costs into capital and recurrent costs. Common programmatic costs (indirect and overhead costs that were shared across various ACF activities) were first calculated as total cost and were apportioned into each major ACF activity category based on direct human resource contribution (ratios of LOE across all ACF activities, weighted based the total person-time contribution assessed for each activity category). Cost data and program operation statistics were collected on a quarterly basis from July 2017 to December 2018 based on program financial documents and interviews with program managers. We evaluated these data against the respective service utilization and program statistics (e.g. number of patients)

both quarterly and over the entire program period (total of 18 operational months). The main outcomes of the analysis were 1) unit costs of key service/activity components of the ACF intervention (calculated based on direct total costs of each discrete service/activity divided by the total number of patients who received the service); and 2) cost per direct program yield (cost per presumptive TB patient identified and per confirmed TB diagnosis). Capital assets were annualized based on the relevant expected life years and discounted at a 3% annual rate. All costs were assessed as economic costs from the health system perspective and reported in 2018 United States Dollars (USD) with cost data in local currencies converted based on the average United Nations operational exchange rates for 2018 [20].

## Cost-effectiveness analysis

Costs and effect estimates were estimated using a multi-stage Markov state-transition model (Fig 1) using monthly time steps. Two stages included 1) a symptom transition model to calculate symptom-based care-seeking probabilities without ACF and 2) a decision-analytic model representing ACF and Passive Case Finding (PCF) as a status quo diagnostic and treatment algorithm. In the symptom transition model (developed using Microsoft Excel), we calibrated the underlying monthly care-seeking probabilities and symptom transition rates to reproduce a population consistent with current TB epidemiology in Zambia for each of three symptom levels: 1) TB-asymptomatic (patients unaware of or without any TB-specific symptoms); 2)

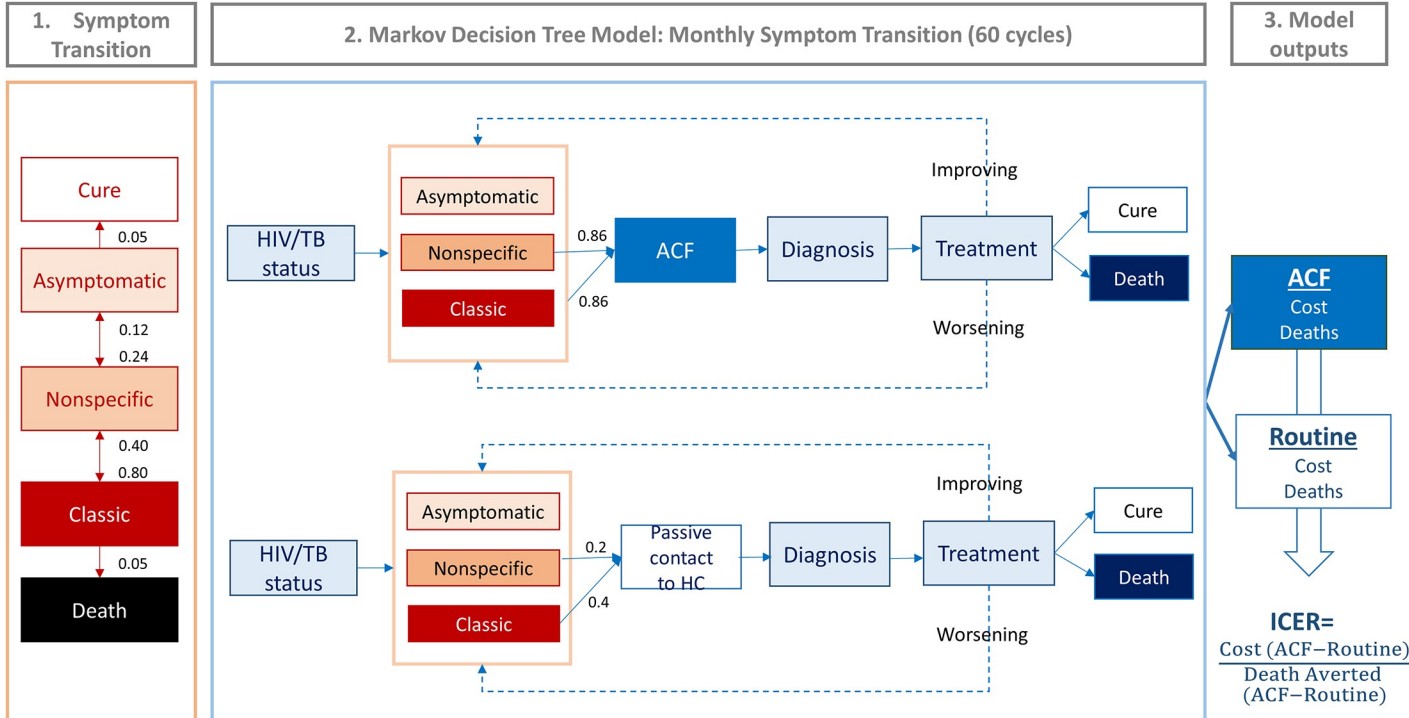

**Fig 1. Conceptual framework for the Markov model.** In our symptom-based care seeking model, we defined three TB-symptom levels—asymptomatic, nonspecific, classic—based on the corresponding probability of diagnostic evaluation for TB. We calculated monthly transition rates between these symptom levels based on three constraints: 1. Probability of progression is 2 times that of regression; 2. Lifetime probability of TB self-cure equals that of death in the absence of treatment (untreated case fatality ratio of 0.5); and 3. Mean duration of asymptomatic period is 9 months. These values (monthly transition rates between symptom levels and monthly probabilities of seeking care) were inputted into a decision tree Markov model which was constructed to reflect the diagnostic algorithm (CXR and Xpert) used for Active Case Finding (ACF) in the Zambia TB REACH program. 100,000 individuals defined by TB/HIV status and symptom level and modeled as having a one-time chance to attend ACF (86% for nonspecific and classic symptom). Those who did not access ACF were modeled as seeking routine care with a monthly probability based on symptom development (20% for nonspecific and 40% for classic symptom) throughout the duration of the analysis. Individuals with untreated TB at the end of each monthly cycle experienced a monthly probability of symptom level transition (progression or regression). More detailed model structure and clinical diagnostic algorithms are described in the supporting information S2 and S3 Tables in S1 File; S2 and S3 Figs in S1 File.

non-TB-specific symptoms (symptoms for which an Xpert test would be ordered, if patients were to present passively to the clinic) and 3) classical TB symptoms (symptoms for which TB treatment would be started empirically, even if Xpert testing were negative). Additional details of this symptom transition model are described in the Supporting Information (S2 and S3 Tables in S1 File and S2 and S3 Figs in S1 File) and elsewhere [21]. Symptom-specific care-seeking probabilities from the symptom transition model were incorporated into the decision-analytic model to estimate the incremental effect (diagnosis and averted TB mortality) of the ACF intervention over the five-year time horizon compared to the status quo. The decision-analytic model, built-in TreeAge Pro 2018 (Williamstown, MA, USA), represented the overall TB care cascade for both passive and active case finding for a simulated population of 100,000 individuals. Epidemiologic parameters for our symptom-transition model were obtained from national estimates and published literature [22, 23–27]. Cost-effectiveness of the CIDRZ ACF operation in a setting with TB epidemiology and economic conditions consistent with national averages was evaluated based on the incremental cost-effectiveness ratio (ICER), calculated as incremental cost per TB death averted over a five-year time horizon (chosen as the minimum interval that might occur between serial intensive ACF campaigns in practice), relative to the status quo (PCF). Complete list of parameters used in the model can be found in Table 1 and S3 Table in S1 File.

## Sensitivity analysis

To test the robustness of our cost-effectiveness estimates, we performed a suite of sensitivity analyses (one-way, three-way, and probabilistic sensitivity analyses) based on the uncertainty estimates of each parameter. Parameters for the three-way sensitivity analysis were selected based on the ranking of the top three most influential parameters identified from one-way sensitivity analysis. For the Probabilistic Sensitivity Analysis (PSA), all model parameter values were randomly sampled over 10,000 Monte Carlo simulations based on pre-specified distributions of each data parameter to generate 95% uncertainty ranges. ICER estimates were evaluated against different willingness-to-pay (WTP) thresholds representing a range of financial and budgetary constraints in Zambia for public health interventions [28].

## Ethical statement

Neither ethical approval nor informed consent was required for this analysis which did not involve human subjects research. Neither patients nor the public were involved in the design, conduct, reporting, or dissemination plans of our research.

## Results

In our observation of program operations (July 2017 to December 2018), the comprehensive ACF program registered and screened 20,386 patients (mean 1,133 per month) and detected 943 new TB cases (mean 52 per month), incurring a total cost of $433,078 (S1 Table in S1 File). Various operational and external factors influenced service volume and unit costs over time, as described in S1 Fig in S1 File. As shown Table 2, unit cost per screening varied between $2 and $6 (mean $3.33) as quarterly service volume varied between 1724 and 6601; unit cost per mCXR-CAD4TB screening varied between $8 and $26 (mean $13.31) as service volume varied between 464 and 6038; and unit cost per Xpert test varied between $10 and $28 (mean $16.34) as service volume varied between 718 and 1179. On the other hand, the number of confirmed TB cases was reasonably constant varying between 151 and 190 per quarter. Overall, the ACF intervention cost an estimated $435 per TB treatment initiated by all diagnostic methods including mCXR-CAD4TB, Xpert, and empiric diagnosis.

**Table 1. Key model parameters.**

| Model Parameter | | Base Value | Distribution | Low Value | High Value | Source |
|---|---|---|---|---|---|---|
| *Disease Epidemiology* | | | | | | |
| People living with Human Immunodeficiency Virus (HIV) prevalence | | 0.113 | Beta | 0.1 | 0.13 | 22 |
| Proportion of Tuberculosis (TB) occurring in HIV+ patients | | 0.59 | Beta | 0.46 | 0.70 | |
| Proportion of TB cases who have not been treated in the past | | 0.95 | Beta | 0.90 | 1.0 | |
| TB prevalence in general population | | 0.00346 | Beta | 0.0029 | 0.0043 | |
| Proportion of MDR+ TB cases who are previously treated [a] | | 0.18 | Beta | 0.14 | 0.22 | |
| Proportion of MDR+ TB cases, in treatment-naive patients | | 0.028 | Beta | 0.025 | 0.031 | |
| *Efficacy of Diagnostic Tests* | | | | | | |
| Sensitivity of chest X-ray | | 0.77 | Beta | 0.70 | 0.90 | 24 |
| Specificity of Xpert | | 0.98 | Beta | 0.97 | 0.99 | 24 |
| Sensitivity of Xpert for High Bacterial Loads [b] | | 0.98 | Beta | 0.97 | 0.99 | 26 |
| Sensitivity of Xpert for Low Bacterial Loads [b] | | 0.68 | Beta | 0.59 | 0.75 | 27 |
| *Monthly Untreated TB Mortality [c] (per 1000 person-years)* | | | | | | |
| HIV positive, Smear positive | | 0.06 | Beta | 0.0408 | 0.0799 | 25 |
| HIV positive, Smear negative | | 0.054 | Beta | 0.0408 | 0.0799 | |
| HIV negative, Smear positive | | 0.021 | Beta | 0.0176 | 0.0288 | |
| HIV negative, Smear negative | | 0.0083 | Beta | 0.0071 | 0.0095 | |
| *Connection to Health Care System* | | | | | | |
| Proportion of (nonspecific/classic) symptomatic patients who attend active case finding program | | 0.86 | Beta | 0.5 | 1.0 | S3 Table in S1 File |
| Monthly probability of a patient passively contacting the health system for TB diagnosis | Asymptomatic | 0 | Beta | 0 | 0 | |
| | Nonspecific | 0.2 | Beta | 0 | 0.5 | |
| | Classic | 0.4 | Beta | 0.2 | 0.6 | |
| *Monthly Symptom Level Transition Rate* | | | | | | |
| Probability of transition from no symptom to cure | | 0.050 | Beta | 0.04 | 0.06 | S3 Table in S1 File |
| Probability of transition from mild symptom to no symptom | | 0.120 | Beta | 0.1 | 0.2 | |
| Probability of transition from no symptom to mild symptom | | 0.240 | Beta | 0.2 | 0.3 | |
| Probability of transition from mild symptom to strong symptom | | 0.800 | Beta | 0.7 | 0.9 | |
| Probability of transition from strong symptom to mild symptom | | 0.400 | Beta | 0.3 | 0.8 | |
| Probability of transition from strong symptom to death | | 0.050 | Beta | 0.4 | 0.6 | |

[a]. While multi-drug resistant (MDR) TB is not a part of standard monitoring indicators of the TB REACH program in Zambia, we incorporated a probability of MDR TB for the Markov state-transition model based on the country estimates.

[b]. "High bacterial load" is defined as TB that, if tested with a single sputum smear under programmatic conditions, would test positive; "Low bacterial load" is defined as TB that, if tested with a single sputum smear under programmatic conditions, would test negative.

[c]. Monthly mortality rate was estimated based on 1-EXP(-annual rate/12 months) from Vassall et al.[25] and WHO Zambia TB country profile (TB case fatality ratio as 31% in 2018)[20].

Our model estimated that the one-off comprehensive ACF intervention implemented to cover 100,000 people in a generalized Zambian adult population (i.e., reflective of the epidemiology and economic conditions of Zambia as a whole) would incrementally diagnose 407 TB cases (7,207 vs. 6,800), at an incremental diagnostic cost of $822,000 (incremental cost of $2,020 per TB patient diagnosed), compared to passive case finding over a five-year horizon. This incremental and early diagnosis of TB patients by the ACF intervention would avert 498 TB deaths (612 vs 1,110) at incremental total health systems cost of $1,110,000 (ICER: $2,284 per death averted). Full cost-effectiveness analysis outcomes are available in Table 3.

**Table 2. Service volumes and unit costs associated with an active case finding program for tuberculosis in Zambia.**

|  | Types of services | Number of beneficiaries | Unit cost | Ranges [a] |
|---|---|---|---|---|
| Cost per activity [b] | Cost per patient screened | 18,662 | $3.33 | ($1.53, $5.79) |
|  | Cost per patient diagnosed by Chest X-ray | 12,679 | $13.31 | ($7.54, $25.62) |
|  | Cost per patient diagnosed by Xpert | 4,511 | $16.34 | ($9.68, $27.89) |
| Cost per yield | Cost per patient treated based on positive Chest X-ray | 359 | $553 | ($314, $1004) |
|  | Cost per patient treated based on positive Xpert | 471 | $755 | ($509, $985) |
|  | Cost per treatment initiated (by all diagnostic method) | 847 | $435 | ($313, $659) |

[a]. Ranges represent the highest and lowest unit cost in a given quarter, accounting for observed variability from quarter to quarter over an 18-month period (S1 Fig and S1 Table in S1 File)

[b]. Cost per activity estimates were used as cost parameters for the Markov state transition model.

Our one-way sensitivity analyses found that the incremental cost-effectiveness of ACF was most sensitive to estimated mortality among people with HIV and smear-positive TB, followed by the monthly probability of passive care-seeking and the unit cost of ACF screening (Fig 2). In three-way sensitivity analysis (Fig 3), ACF was most cost-effective when HIV/TB mortality was higher, symptom-associated passive care-seeking was less common, and the unit cost of ACF screening was lowest. Our probabilistic sensitivity analysis demonstrated that the likelihood of ACF cost-effectiveness substantially improved when evaluated over longer time horizons (Fig 4). If we assume a five-year time horizon, the probability of the cost-effectiveness of the ACF intervention was > 90% when policymakers in Zambia are willing to consider a value of $4,000 or higher for each death averted by the intervention. This would fall to 70% when considering only a one-year time horizon (Fig 4). In a longer-term assessment (five years), our model estimated that incremental diagnoses made by a one-off intensive ACF intervention may fall in the latter years as individuals with active TB subsequently seek routine care. Nevertheless, a substantial number of deaths were averted because individuals with TB were diagnosed earlier through ACF; these incremental benefits were mostly achieved within the first two years of ACF implementation (Fig 4). In settings where patients are more likely to passively seek care upon symptom onset (e.g., increase in the probability of symptom-associated care-seeking), the same ACF intervention would avert less than half as many TB deaths (reduced from 750 to 300 per 100,000 population) and would take a longer time to realize the full health benefit (16 vs. 28 months) as compared to settings with lower care-seeking probabilities (S4 Fig in S1 File). Depending on the costs of ACF and HIV/TB mortality levels, estimated ICERs varied by a factor of three (Fig 3).

## Discussion

In this study, we evaluated the costs and cost-effectiveness of a one-time intensive ACF intervention in Zambia that included combined community and facility-based screening program

**Table 3. Five-year epidemiological and economic outcomes of 100,000 individuals exposed to status quo or active tuberculosis case finding.**

| Diagnostic Strategy | TB Diagnoses (95% uncertainty interval) | TB Deaths (95% uncertainty interval) | Total Diagnostic and Treatment Cost (95% uncertainty interval) | ICER (USD/Death Averted) |
|---|---|---|---|---|
| Active Case Finding (ACF) | 7,207 (6,342, 8,488) | 612 (405, 1111) | $3,897,000 ($3,314,000, $4,932,000) |  |
| Passive Case Finding (Status quo) | 6,800 (5,912, 8,090) | 1,110 (700, 1994) | $2,787,000 ($2,339,000, $3,610,000) |  |
| Incremental (ACF—Status quo) | 407 (397, 430) | -498 (-296, -883) | $1,110000 ($975,000, $1,322,000) | $2,284 ($1,497, $3,298) |

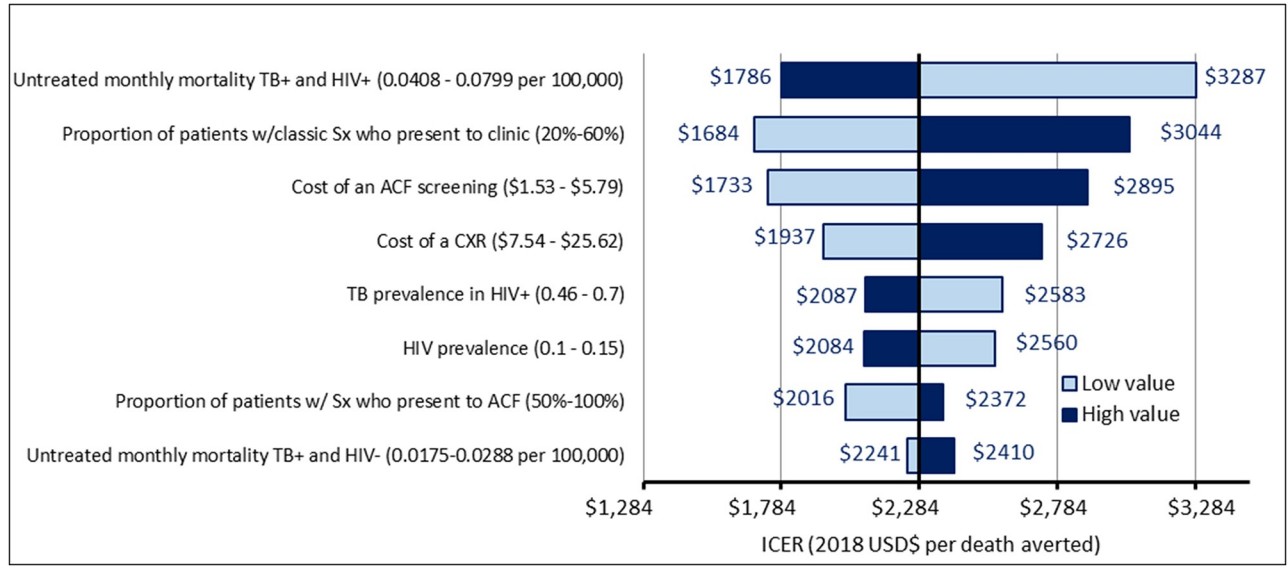

**Fig 2. One-way sensitivity analysis of the cost-effectiveness of active tuberculosis case finding over five years (2018–2022) in Zambia.** The parameters shown had the greatest quantitative influence on the incremental cost-effectiveness of ACF relative to routine care in one-way sensitivity analysis. Bars show the incremental cost-effectiveness (2018 US dollars per death averted averted) of ACF relative to routine care under the high value (dark blue bar) and low value (light blue bar) of the parameter in question, holding all other parameters constant. For example, when the untreated monthly mortality of HIV/TB (HIV positive and smear positive) is low, the ICER increased from the base value of $2,284 to $3,287, suggesting that the ACF intervention is less cost-effective compared to the routine program. The vertical line corresponds to the reference scenario (values as in Table 1, corresponding to $2,284 per death averted).

| Low HIV/TB mortality rate (0.0408 per 1000 person years) | | | | | | ICER (USD/Death averted) |
|---|---|---|---|---|---|---|
| | Proportion with classic symptoms who seek care per month | | | | | $6,000-$6,999 |
| | 0.2 | 0.3 | 0.4 | 0.5 | 0.6 | $5,000-$5,999 |
| Cost per ACF screening — $5.79 | $2,820 | $3,465 | $4,208 | $5,703 | $6,090 | $4,000-$4,999 |
| Cost per ACF screening — $3.33 | $2,209 | $2,668 | $3,198 | $3,814 | $4,538 | $3,000-$3,999 |
| Cost per ACF screening — $1.53 | $1,765 | $2,090 | $2,464 | $2,899 | $3,411 | $2,000-$2,999 |

| Base case HIV/TB mortality rate (0.0598 per 1000 person years) | | | | | | $1,000-$1,999 |
|---|---|---|---|---|---|---|
| | Proportion with classic symptoms who seek care per month | | | | | |
| | 0.2 | 0.3 | 0.4 | 0.5 | 0.6 | |
| Cost per ACF screening — $5.79 | $2,077 | $2,465 | $2,900 | $3,390 | $3,944 | |
| Cost per ACF screening — $3.33 | $1,646 | $1,919 | $2,284 | $2,568 | $2,957 | |
| Cost per ACF screening — $1.53 | $1,332 | $1,522 | $1,733 | $1,971 | $2,240 | |

| High HIV/TB mortality rate (0.0799 per 1000 person years) | | | | | |
|---|---|---|---|---|---|
| | Proportion with classic symptoms who seek care per month | | | | |
| | 0.2 | 0.3 | 0.4 | 0.5 | 0.6 |
| Cost per ACF screening — $5.79 | $1,695 | $1,959 | $2,251 | $2,574 | $2,933 |
| Cost per ACF screening — $3.33 | $1,356 | $1,540 | $1,741 | $1,964 | $2,212 |
| Cost per ACF screening — $1.53 | $1,111 | $1,235 | $1,371 | $1,521 | $1,688 |

**Fig 3. Three-way sensitivity analysis of the cost-effectiveness of active tuberculosis case finding over five years (2018–2022) in Zambia.** This heat map displays the incremental cost-effectiveness of active TB case finding (ACF) compared to routine care, in units of cost per death averted. Each panel corresponds to a relative HIV/TB mortality rate (0.75 times base case, base case, 1.25 times of base case), with each column representing a different monthly probability of routine care-seeking behavior and each row depicting a different level of unit cost per ACF screening (as described as a measure of productivity level in ACF screening). The base case in our analysis had a HIV/TB mortality rate of 0.0598 per 1000 person years, 40% of individuals with classic symptoms seeking care to the clinic per month and cost per ACF screening of $3.33, resulting in an ICER of $2,284 per death averted.

using mCXR-CAD4TB followed by Xpert testing, and a linkage-to-care program to facilitate TB diagnosis and treatment initiation. In an average Zambian setting, represented by HIV/TB burden and care-seeking patterns similar to the national averages, we expect similar ACF interventions can avert a TB death at a cost of $2,284 (95% uncertainty: $1,497 to $3,298). This would translate to $99 per year of life saved (assuming average life expectancy of Zambian TB patients to be 40 years, discounted at 3% at the time of diagnosis), which is likely below Zambia's country-specific cost-effectiveness threshold for health interventions (e.g., $68-$768 per quality-adjusted life year gained) [29], when considering earlier and incremental case detection by ACF interventions can also Our findings suggest that a comprehensive ACF intervention that can effectively close gaps between steps of TB screening, diagnosis, and treatment initiation may have greater value in settings with higher HIV/TB burdens and high patient care-seeking barriers compared to settings with lower HIV/TB disease burdens (e.g., ICER: $4,284 in Cambodia vs $2,284 in Zambia) [21]. However, the cost-effectiveness of any ACF intervention will vary depending on 1) operational factors influencing the costs of key intervention components (e.g., screening and use of mCXR); 2) level of patient symptom-associated care seeking—an area in which empiric data are lacking; and 3) the duration over which patient-relevant outcomes (e.g. TB case detection and death) are assessed.

Earlier studies evaluating the cost-effectiveness of ACF interventions largely focused on examining programmatic components and strategies (e.g. use of more accurate screening diagnostic tools such as Xpert) [14, 30–32]. While many of these analyses suggest ACF interventions can be highly cost-effective in many different settings, they do not consider supply (e.g. ACF intervention capacity, operational and implementation efficiency that may determine the costs of the intervention) and demand (e.g. patient symptom-associated care-seeking behavior) factors that may be important determinants of allocative efficiency. In this analysis, costs of key programmatic components of the ACF intervention (e.g. patient screening and mCXR) varied considerably depending on patient volumes; these may be contextualized to an intervention's implementation and operational conditions (S1 Fig in S1 File). Unit costs were likely highest during early (with low operational efficiency) and late intervention periods (as the CIDRZ ACF project operated in a single operational catchment area, the number of individuals eligible for screening in the area may have fallen). Measuring the extent and trajectory of service volume and unit cost variation over time (as the product of HIV/TB prevalence, patients' care-seeking and diagnostic sensitivity under both routine and ACF interventions) can provide important insight in evaluating key drivers of cost and cost-effectiveness in specific contexts.

Using an earlier modeling framework [21], we also explored the mechanisms of patient-level factors (symptom-associated care-seeking behavior) in determining the value of ACF interventions in high HIV burden settings. As shown in S4 Fig in S1 File, the timing and magnitude of incremental diagnosis and TB deaths averted differed by rates of symptom-associated care-seeking in the absence of ACF. In settings where patients seek TB care less readily, incremental health benefits (case detection and death aversion) achieved by ACF were twice as large and fast compared to settings where patients sought care more rapidly. Subsequently, if a shorter analytic time frame was used (less than 2 years) to assess the cost-effectiveness of the ACF intervention, it would not fully capture the future benefits of early and incremental case detection (Fig 4). As with an earlier modeling study [33], our results reiterate the importance of considering a longer analytic time frame to assess the potential health population benefits of early and incremental case detection achieved by ACF interventions. Furthermore, our study findings also highlight the significance of the relationship between TB natural history and care-seeking behavior in determining the cost-effectiveness of a one-time ACF intervention. As the factors and causality of patient care-seeking patterns will vary from one setting to

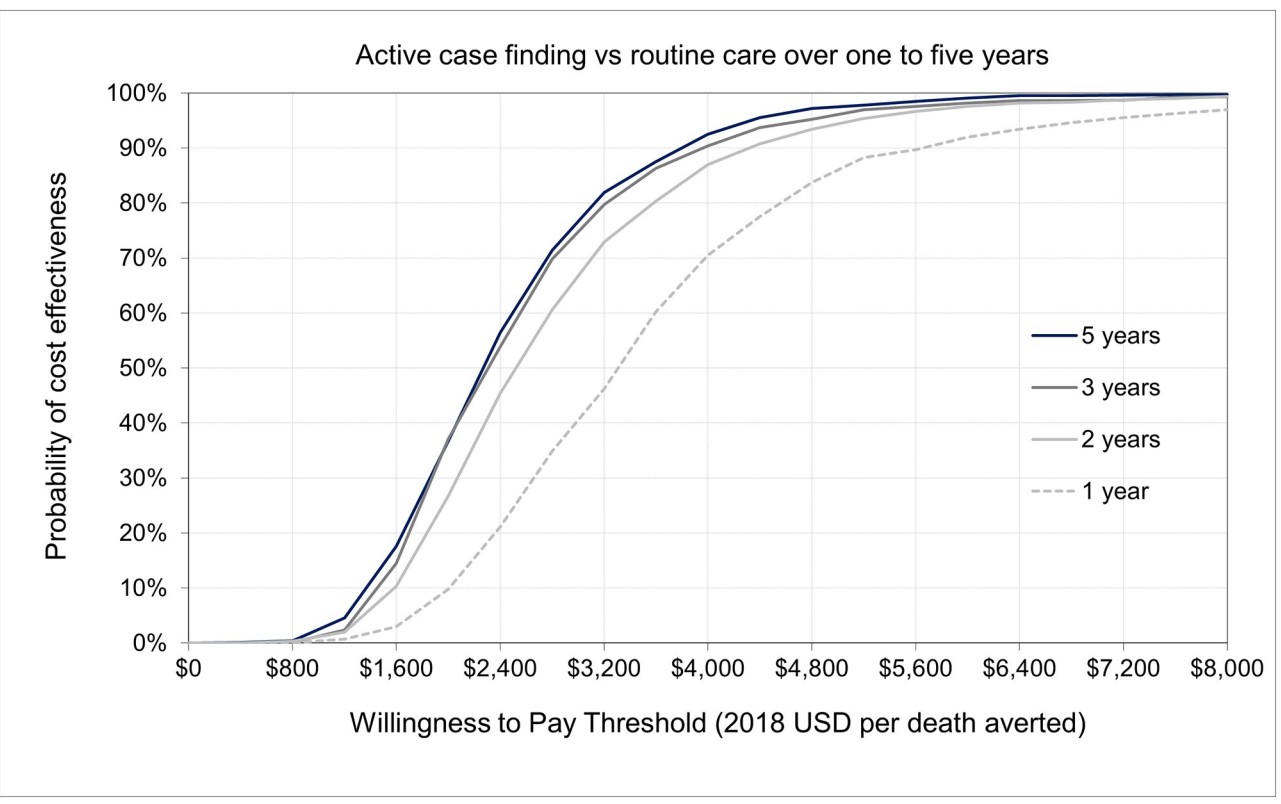

**Fig 4. Cost effectiveness acceptability curve of an active tuberculosis case finding intervention compared to routine care, over one to five years (2018–2022) in Zambia.** In this figure, the horizontal axis denotes the willingness to pay (WTP) per death averted (incremental cost-effectiveness ratio, ICER), and the vertical axis indicates the probability of ACF being cost-effective based on the proportion of simulations in which the comparison of the ACF intervention to the routine program falls below the WTP threshold. Costs are expressed in 2018 US dollars. At a WTP threshold of $4,800 per TB death averted (about three times Zambia's GDP per capita in 2018), 97% of simulations suggest that the ACF intervention will be cost-effective compared to the routine program in Zambia over 5 years. This percentage declines to 95% over a three-year horizon, 93% over two years, and 83% in the first year of implementation.

another, understanding the local contexts in which TB care cascade gaps influence patient care-seeking patterns will help optimize the design and implementation strategy of ACF interventions.

Our study findings illustrate two distinctive implications for the dynamics of cost-effectiveness of ACF over time and place. First, S4 Fig in S1 File demonstrates how the cost-effectiveness may change over time among the same (fixed) population in a given catchment area, driven by their symptom progression and associated care-seeking. This suggests that cost-effectiveness may differ by stage of implementation. For example, in the initial stage of program implementation, HIV/TB prevalence, care-seeking patterns, and associated symptom progression may be key drivers of cost-effectiveness, while operational efficiency may have greater influence on cost-effectiveness at later stages (as prevalence decreases and service volume increases over time). Our study estimated that incremental cost-effectiveness might fall about 30% (i.e. improved cost-effectiveness) from $3,200 in the first year to $2,284 in the fifth year (Fig 4). On the other hand, Fig 3 demonstrates to what extent the cost-effectiveness of ACF may differ across different regions and populations, driven by underlying heterogeneity in risks or behaviors within those populations. For example, another study that evaluated ACF in Cambodia estimated cost-effectiveness at $5,300 per death averted [30], more than two times greater than estimated here ($2,284 per death averted). Since HIV/TB mortality and

symptom-associated passive care seeking may not be easily modifiable, efforts to increase ACF service utilization in high prevalence areas through demand creation among key populations (e.g. people living with HIV or household contacts of people diagnosed with active TB) and/or optimization of intervention duration may be key strategies for program managers to improve the cost-effectiveness of ACF programs.

As with any modeling analysis, our study has some limitations. First, we calibrated care-seeking behavior to national estimates of prevalence and symptom duration but did not model underlying factors affecting patients' care-seeking behavior that may vary within and across countries [34, 35]. These factors may include a wide range of health systems and patient-level barriers [36, 37] that may be difficult to model without empiric data in the context of TB case finding. Moreover, the extent of care seeking and the value of ACF interventions can also be influenced by other health system factors (e.g. quality of care) and other patient factors (e.g. community awareness of TB and available services). Given the importance of these factors in determining the value of ACF interventions, we encourage future observational and modeling studies to investigate how and to what extent different health system and patient factors might impact the incremental benefits and costs of ACF. Second, we assessed the costs and cost-effectiveness of the CIDRZ ACF intervention as a combined package, including multiple programmatic components that were deployed in both the community and the facility. Our analysis of effectiveness was limited to individuals diagnosed by the intensified case-finding and thus may underestimate the value of this intervention in encouraging individuals to also seek routine care for TB diagnosis. For example, a fast track point operating within GPHC was found to be an important entry point for TB diagnosis [38]; if continued, such interventions could—with minimal ongoing cost and effort—help to encourage ongoing care-seeking for TB diagnosis through routine services as well. Third, our empiric cost estimates were assessed using a top-down method only and were limited to one semi-urban health clinic in Zambia. Economic cost estimates of health services may vary depending on settings in which the services are deployed, methods used to evaluate costs, and how costs associated with implementation are accounted for in the analysis [39, 40]. While our findings may not generalize to epidemiological and health system contexts that are very different from our study, costs assessed at various points provide empiric uncertainty estimates of the cost of ACF and can be contextualized to different levels of operational and implementation efficiencies. Lastly, we did not include the effects of ACF on TB transmission; therefore, longer-term effects of ACF on TB incidence were not assessed. Instead, we focused on the trajectory of service output and impact of early case detection on TB death up to five years and demonstrate that the overall cost-effectiveness was robust to a range of sensitivity analyses. Likewise, if the effects of ACF on TB transmission were included, it would further strengthen the economic and epidemiologic case of ACF interventions in high HIV/TB burden settings [33].

In conclusion, our study demonstrates that a comprehensive ACF intervention model explored by CIDRZ can be cost-effective in populations representative of the epidemiological and economic conditions in Zambia—and likely in other contexts where HIV/TB burden is high. The value of ACF may be optimized in settings where HIV/TB mortality is high, existing care seeking is infrequent, and when ACF interventions can optimize patient screening by achieving operational efficiency. As these conditions may dynamically change throughout program implementation, individual ACF programs should carefully assess and actively monitor these indicators to identify the optimal timing and duration of operations. Better understanding of patients' symptom-associated passive care-seeking patterns can also help improve the operational focus of ACF intervention and optimization of resource allocation for TB diagnosis and linkage to care in resource-limited settings.

## Supporting information

**S1 File.**
(DOCX)

## Acknowledgments

We would like to thank the data collection team at the University of Zambia (Adam Silumbwe, Maio Bulawayo, Acklas Phiri, Tikurirekuti), and team members at CIDRZ and George Health Clinic in Lusaka who oversaw and conducted the day-to-day operations of the comprehensive active case finding for TB intervention.

**Disclaimer**: The findings and conclusions presented in this report are those of the authors and do not necessarily represent the official position of the authors' affiliated institutions.

## Author Contributions

**Conceptualization:** Youngji Jo, Karl Johnson, David Dowdy, Hojoon Sohn.

**Data curation:** Mary Kagujje.

**Formal analysis:** Youngji Jo.

**Funding acquisition:** Hojoon Sohn.

**Investigation:** David Dowdy, Monde Muyoyeta, Hojoon Sohn.

**Methodology:** Youngji Jo, Karl Johnson, David Dowdy, Hojoon Sohn.

**Project administration:** Mary Kagujje, Peter Hangoma, Lophina Chiliukutu.

**Resources:** Mary Kagujje, Peter Hangoma, Lophina Chiliukutu, Monde Muyoyeta, Hojoon Sohn.

**Software:** Karl Johnson, Hojoon Sohn.

**Supervision:** Mary Kagujje, David Dowdy, Monde Muyoyeta, Hojoon Sohn.

**Visualization:** Youngji Jo.

**Writing – original draft:** Youngji Jo.

**Writing – review & editing:** Mary Kagujje, Karl Johnson, David Dowdy, Peter Hangoma, Monde Muyoyeta, Hojoon Sohn.

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
