## [Decision Letter · Decision Letter 0]

20 May 2021

PONE-D-21-10534

Costs and Cost-Effectiveness of a Comprehensive Tuberculosis Case Finding Strategy in Zambia

PLOS ONE

Dear Dr. Sohn,

Thank you for submitting your manuscript to PLOS ONE. After careful consideration, we feel that it has merit but does not fully meet PLOS ONE’s publication criteria as it currently stands. Therefore, we invite you to submit a revised version of the manuscript that addresses the points raised during the review process.

We look forward to receiving your revised manuscript.

Kind regards,

Kevin Schwartzman

Academic Editor

PLOS ONE

Additional Editor Comments:

Thank you for submitting your manuscript to PLOS ONE. Both reviewers have identified some areas for improvement and clarification. Please address these carefully when preparing your revised manuscript, including suitable proofreading and editing.

Journal Requirements:

Reviewers' comments:

Reviewer's Responses to Questions

**Comments to the Author**

1. Is the manuscript technically sound, and do the data support the conclusions?

Reviewer #1: Yes

Reviewer #2: Yes

2. Has the statistical analysis been performed appropriately and rigorously? 

Reviewer #1: Yes

Reviewer #2: Yes

3. Have the authors made all data underlying the findings in their manuscript fully available?

Reviewer #1: Yes

Reviewer #2: Yes

4. Is the manuscript presented in an intelligible fashion and written in standard English?

Reviewer #1: Yes

Reviewer #2: Yes

5. Review Comments to the Author

Reviewer #1: This paper describes a cost-effectiveness analysis of active case finding for tuberculosis, in a Zambian setting. This is an excellent paper that fills a real gap in the literature – the analysis is high-quality and the writing is very clear.

Minor comments

Line 105: typo ‘available’

Line 102: It would be useful to clarify ‘other departments’ here – was the target audience for the facility-based case finding restricted to people at higher risk for TB (ie. people with HIV), or did it include everybody attending the health clinic? How did the facility-based active case finding differ from ‘intensified case finding’ or a symptom screen at the front door?

Line 109: A bit more detail on methods for cost data collection would be useful. For example, was this a full or incremental costing? How was resource use measured or estimated - did data collection include observation, interviews, records review, or other methods? How were shared costs or overhead costs allocated across cost centres?

Appendix 1: Your description of the impact of factors such as service volume and time on unit costs is great. It might be useful to separate the graphs to show facility-based and community-based ACF separately, as they are quite different interventions and operated at different times so would probably be useful for readers to see the different costs. Can you also clarify how you have a unit cost for CXR during the period the Xray van was broken down (Jul-Dec 2018)?

Line 138: possible typo, I think you mean “Epidemiologic parameters for our symptom-transition model…”?

Line 175: typo ‘effectiveness’

Line 177: You mention the DALYs averted from the intervention in your results, but this is not in tables or in the discussion. It also looks like the only DALYs you include are related to YLLs, and you don’t include any from YLDs – is this right? If your main outcome is deaths, and not DALYs, it might not be necessary to include this sentence on DALYs in the results, especially as it is highly conservative as doesn’t include any reduction in disability through treatment of TB.

Line 206: I think you need to say a bit more here to justify why your incremental cost/death averted would be considered ‘cost-effective’. Most ‘universal’ thresholds use cost per DALY averted, not per death averted. Do you have a working threshold for cost-effectiveness per death averted, or can you give some examples of the cost/death averted for funded interventions?

Line 206: typo ‘expanded’

Line 236: typo ‘likely to seek care passively’

Discussion section: Did you capture any information on changes to passive case finding in the context of the community TB awareness events etc? Would it be worth mentioning these possible broader contextual impacts of ACF schemes if they have a knock-on effect of (for example) increased care-seeking behaviour in the general population?

Reviewer #2: General Comments:

Jo and colleagues evaluate the costs and cost-effectiveness of a large active TB case-finding initiative carried out in Zambia in 2017-18 using a multi-stage Markov transition model. The analysis compared conventional facility-based passive case-finding to facility- plus community-based TB screening using symptoms and automated digital chest radiography plus GeneXpert testing with linkage to TB treatment. The topic is important because a growing body of evidence suggests that active case finding is an effective and likely an essential strategy to be added to passive case finding in order to improve TB control and achieve TB elimination. However, there is a scarcity of information relevant to implementing these programs in real-world settings.

The underlying ACF program increased the number of TB cases detected by about 5%, at a cost of $435 per new treatment, and reduced deaths by almost 50%, at a cost of $2284 per death; these represent an excellent return on investment based on prior studies. The authors highlight several key implications of their analysis. First, their estimates of the unit costs and overall cost-effectiveness of the active case-finding program appeared to be highly sensitive to the available public health capacity and efficiency of implementation in a particular setting (“supply” factors), as well as to the volume of undiagnosed and non-care-seeking TB patients (“demand” factors). Second, similar to previous studies, they found that the benefits of active case finding are likely to accrue over a more extended time period than is commonly realized – at least five years. Both these considerations are likely to be important as countries consider where and how they will or will not adopt such programs and evaluate initial implementation.

Major Comments:

The analyses are well-presented and the figures and tables are very accessible to the reader. I have a few clarifying questions, as well as some questions about how these findings are packaged and presented to policymakers, as this is presumably one of the expected outcomes of publication.

Could the authors define what they mean by “generalized Zambian settings,” as compared with the settings of the CIDRZ ACF project? Are there contextual differences between them, or does this simply refer to nationwide scale-up?

The manuscript presents the differences in unit costs per process measure between different time periods, but how variable were the costs per outcome (i.e. per TB diagnosis, per TB death) by quarter?

Which of the factors influencing the temporal variations in cost are likely to be most easily modifiable? This information is contained in Figure 3, but it might be worth expanding on since HIV/TB mortality and symptom-associated passive care seeking may not be modifiable in ways that make ACF more cost-effective.

While interesting, I wonder if the insights into the important effects of care-seeking behaviors on ACF cost-effectiveness tell the whole story. For example, the model implies that active case finding will be most cost-effective in settings where passive care-seeking behaviors and/or the efficiency of facility-based care is lowest. I imagine that this may commonly occur in facilities where the quality of care and service are poor, and often the low quality of care may also be well-known in the community. Yet, the treatment outcomes of patients identified during active case-finding may also depend on the quality of care in these facilities. If the goal is high rates of treatment coverage and cure, this kind of analysis may not capture these complexities. The authors hint at this in their call for more empiric data across the diagnosis and treatment cascade but I wonder if this specific aspect might be highlighted since the influence of care-seeking behaviors is such a salient finding of this analysis.

In the last paragraph of the Results (Lines 189-198) and in the Discussion (Lines 227-230), the analysis uses temporal differences in cost and cost-effectiveness to generalize about such differences between settings. This seems potentially problematic since the word “settings” implies differences defined on factors other than time (e.g. geography). While time-based comparisons yields insights about how care can be delivered more efficiently to a fixed population (assumed stable, at least over the short term), comparisons between geographic settings should arguably account for selection differences between populations and for underlying heterogeneity in risks or behaviors within those populations, especially to the extent that the boundaries used to define populations geographically are arbitrary yet influence estimates of cost-effectiveness (i.e. different boundaries lead to different conclusions).

Minor Comments:

The manuscript would benefit from a close reading for grammar and usage to improve its readability and clarity.

Line 104: It is implied but not directly stated that specimens were collected for abnormal symptoms or abnormal mCXR; a little more detail on the ACF algorithm at this point in the Methods would be helpful for clarification.

Table 1: The proportion of TB occurring in HIV patients is estimated at 59% (range 46-70%) is justified by an unpublished manuscript (Reference 21). While the Introduction cites similar numbers from the WHO 2020 TB control report, and certainly this estimate reflects case notification data, recent prevalence studies seem to suggest that the contribution of HIV-patients to TB may be lower in active case finding settings, perhaps because of the success of HIV programs in increasing HIV status awareness, and linkage to care which includes TB screening. See for example N. Kapata et al. PLOS ONE (2016).

6. PLOS authors have the option to publish the peer review history of their article (what does this mean?). If published, this will include your full peer review and any attached files.

Reviewer #1: No

Reviewer #2: No

---

## [Author Response · Author response to Decision Letter 0]

23 Jul 2021

July 17, 2021

Responses to the reviewers comments provided for PONE-D-21-10534

Manuscript Title: Costs and Cost-Effectiveness of a Comprehensive Tuberculosis Case Finding Strategy in Zambia

Dear Editors and Editorial Team members,

We are grateful for the opportunity to revise and submit an improved manuscript, reflecting valuable feedback from the editors and reviewers. Specific responses to the editorial and reviewers’ comments are provided in blue, with references (with line numbers) to specific revisions made in the main manuscript (provided with italicized text with double quotes). A track change version of the revised manuscript and supplementing documents are uploaded with ‘R1’ designation. 

We believe that the editor and reviewers’ feedback have resulted in a much improved manuscript and hope the editorial team and the reviewers agree. 

Thank you. 

Sincerely,

Hojoon Sohn, PhD MPH 

Department of Epidemiology

Johns Hopkins Bloomberg of Public Health

615 N. Wolfe Street E. 6039

Baltimore, MD, 21205

United States of America

hsohn6@jhmi.edu

+1-443-517-8145

Additional Editor Comments:

Thank you for submitting your manuscript to PLOS ONE. Both reviewers have identified some areas for improvement and clarification. Please address these carefully when preparing your revised manuscript, including suitable proofreading and editing.

Journal Requirements:

and

Thank you for this information. We have reviewed and updated the manuscript following the PLOS ONE styling. 

We’ve revised the text describing the grant information to match the ‘funding information as well as ‘financial disclosure’ sections. 

We have added the captions for the Supporting Information at the end of the manuscript. 

5. Review Comments to the Author

Reviewer #1: This paper describes a cost-effectiveness analysis of active case finding for tuberculosis, in a Zambian setting. This is an excellent paper that fills a real gap in the literature – the analysis is high-quality and the writing is very clear.

Minor comments

Line 105: typo ‘available’

Line 175: typo ‘effectiveness’

Line 206: typo ‘expanded’

Line 236: typo ‘likely to seek care passively’

Line 138: possible typo, I think you mean “Epidemiologic parameters for our symptom-transition model…”?

Thank you for noting these typos. We have corrected typos throughout the manuscript, including those mentioned above. 

Line 102: It would be useful to clarify ‘other departments’ here – was the target audience for the facility-based case finding restricted to people at higher risk for TB (ie. people with HIV), or did it include everybody attending the health clinic? How did the facility-based active case finding differ from ‘intensified case finding’ or a symptom screen at the front door?

Thank you for raising these questions. The target population for the facility-based case finding was all patients visiting the George primary health care center (GPHC) with either symptoms relating to tuberculosis or people living with HIV (PLHIV) who were making routine care visits to the ART clinic of GPHC. The facility-based team operated from an open access tent in the common grounds of GPHC to facilitate linkage (i.e. conducting initial symptom screening and patient referral for sputum collection and testing) to TB care for those accessing the open access tent. The main difference between facility-based case finding and teams that conducted the door-to-door screening was the location of operation (facility vs. community). 

To further clarify the roles of facility-based case finding and what constitutes ‘other departments’, we have revised the text as the following (line 100-105): 

“The facility-based team operated out of an open-access tent at GPHC where four trained staff members conducted TB symptom screening and patient referral to TB services (including treatment initiation). In addition to screening people presenting (or referred) to the tent, the facility-based team also made regular visits to the antiretroviral therapy (ART) clinic and other departments (e.g. maternal health or general outpatient clinics) at GPHC to identify patients with TB symptoms and/or those patients otherwise indicated for TB screening (e.g., people living with HIV).” 

Line 109: A bit more detail on methods for cost data collection would be useful. For example, was this a full or incremental costing? How was resource use measured or estimated - did data collection include observation, interviews, records review, or other methods? How were shared costs or overhead costs allocated across cost centres?

Thank you for this comment. For our empiric cost analysis, we primarily focused on the costs of ACF intervention service delivery, up to the point of treatment initiation, as incremental program to the routine care. Costs of the ACF intervention were assessed primarily based on our earlier work (also cited in the manuscript – reference #19)[1] but we have provided additional details on the methods used for data collection for resource use and apportionment of common costs across key ACF activities as following (lines 115-122): 

“Human resource costs for each activity component were estimated based on their estimated level of effort (LOE), approximated as proportional time allocation (%) of their full-time work spent on each activity during program operation. Estimated LOE was assessed periodically (each quarter) using a workload survey. For the costs of goods, equipment, and services, we divided direct costs into capital and recurrent costs. Common programmatic costs (indirect and overhead costs that are shared across various ACF activities) were first calculated as total cost and were the apportioned into major ACF activity categories based on direct human resource contribution (ratios of LOE across all ACF activities, weighted based the total person-time contribution assessed for each activity category).” 

In the cost-effectiveness analysis, we accounted for costs of routine care (status quo) with some of these costs empirically assessed from our costing study (routine symptom screening and facility based Xpert testing for which testing capacity was shared with the ACF program). Other routine care costs (e.g. costs of smear and treatment) were extrapolated from published estimates.[2, 3] 

Appendix 1: Your description of the impact of factors such as service volume and time on unit costs is great. It might be useful to separate the graphs to show facility-based and community-based ACF separately, as they are quite different interventions and operated at different times so would probably be useful for readers to see the different costs. Can you also clarify how you have a unit cost for CXR during the period the Xray van was broken down (Jul-Dec 2018)?

In our analyses, we considered community and facility-based components of the ACF program as a single intervention package to be compared to the status quo (absence of ACF intervention). This was primarily done due to limitations in collecting activity-based data as would be needed to assess costs using a bottom-up approach. Therefore, we used top-down approach to conduct cost analysis and could not distinguish costs of activities performed at the facility and in the community. This limitation extends to calculating costs of X-ray. While the truck with the mobile X-ray unit installed was not operational during the months between July and December of 2018, the truck was parked at GPHC (facility) to provide X-ray services for those patients being screened by the facility-based teams and those who were being referred to the clinic from the community-based activities (e.g. drama clubs). 

Line 177: You mention the DALYs averted from the intervention in your results, but this is not in tables or in the discussion. It also looks like the only DALYs you include are related to YLLs, and you don’t include any from YLDs – is this right? If your main outcome is deaths, and not DALYs, it might not be necessary to include this sentence on DALYs in the results, especially as it is highly conservative as doesn’t include any reduction in disability through treatment of TB.

Thank you for noting this. You are correct that our main cost-effectiveness outcome was assessed based on incremental deaths averted by the ACF intervention. Therefore, we have removed the references to DALYs in the Results section accordingly. Instead, we have included a statement in the Discussion section (with corrected notion of life years saved vs. DALYs) so that readers can contextualize our ICER estimates to effectiveness measures (e.g. life years saved) compared to Zambia’s country-specific cost-effectiveness threshold for health interventions (e.g., $68-$768 per quality-adjusted life year gained)[4]. Revisions in the text reference here are shown in our response to your next comment below. 

Line 206: I think you need to say a bit more here to justify why your incremental cost/death averted would be considered ‘cost-effective’. Most ‘universal’ thresholds use cost per DALY averted, not per death averted. Do you have a working threshold for cost-effectiveness per death averted, or can you give some examples of the cost/death averted for funded interventions?

Thank you for your suggestion. We have now added language that would help readers convert a cost per death averted into a cost per year of life saved. This latter quantity is still a conservative estimate of cost per DALY averted or cost per QALY gained (as we would expect the intervention to also avert TB-related morbidity, in the same vein as the Reviewer’s comment about YLDs vs YLLs above). But given that the cost per year of life saved is below most standard estimates of Zambia’s willingness to pay, we would expect the cost per DALY averted (or QALY gained) to be even more favorable to the intervention.

We have now added the following text, benchmarked to country-specific estimates of cost-effectiveness in Zambia, to help clarify these concerns (lines 211-217):

“In an average Zambian setting, represented by HIV/TB burden and care-seeking patterns similar to the national averages, we expect similar ACF interventions can avert a TB death at a cost of $2,284 (95% uncertainty range: $1,497 to $3,298). This would translate to $99 per year of life saved (assuming average life expectancy of Zambian TB patients to be 40 years, discounted at 3% at the time of diagnosis), which is likely below Zambia’s country-specific cost-effectiveness threshold for health interventions (e.g., $68-$768 per quality-adjusted life year gained), especially when considering earlier and incremental case detection by ACF interventions can also avert TB morbidity in addition to mortality.”

Discussion section: Did you capture any information on changes to passive case finding in the context of the community TB awareness events etc? Would it be worth mentioning these possible broader contextual impacts of ACF schemes if they have a knock-on effect of (for example) increased care-seeking behaviour in the general population?

Thank you for the valuable comment. Our study was limited to assessing costs and cost-effectiveness of the CIDRZ ACF intervention; therefore, we did not capture or evaluate changes to passive case finding the context of the community TB awareness events conducted by the ACF project. While it is difficult to ascertain whether community-based activities had a direct effect on changing care-seeking behaviors, our co-authors have previously demonstrated that a fast track point – an access point designed to facilitate linkage to care for both the regular patients visiting the clinic and those being referred from the community – was an important access point for people ultimately diagnosed with bacteriologically confirmed TB (see Table 2 of Kagujje et al., 2020, below).[5]

Area of ACF activity Bacteriologically confirmed TB (%, out of 563 identified by the entire project)

Community 48 (8.5)

Overall facility 515 (91.5)

ART 49 (8.7)

MCH 3 (0.5)

OPD 232 (41.2)

Fast track 214 (38.0)

TB clinic 2 (0.4)

VCT 9 (1.6)

We have subsequently updated our discussion to highlight that ACF may also have positive “knock-on” effects of encouraging people to seek care passively as well (lines 285 – 290):

“Our analysis of effectiveness was limited to individuals diagnosed by active case-finding and thus may underestimate the value of this intervention in encouraging individuals to also seek routine care for TB diagnosis. For example, a fast track point operating within GPHC was found to be an important entry point for TB diagnosis[5]; if continued, such interventions could – with minimal ongoing cost and effort – help to encourage ongoing care-seeking for TB diagnosis through routine services as well.”

Reviewer #2: General Comments:

Jo and colleagues evaluate the costs and cost-effectiveness of a large active TB case-finding initiative carried out in Zambia in 2017-18 using a multi-stage Markov transition model. The analysis compared conventional facility-based passive case-finding to facility- plus community-based TB screening using symptoms and automated digital chest radiography plus GeneXpert testing with linkage to TB treatment. The topic is important because a growing body of evidence suggests that active case finding is an effective and likely an essential strategy to be added to passive case finding in order to improve TB control and achieve TB elimination. However, there is a scarcity of information relevant to implementing these programs in real-world settings.

The underlying ACF program increased the number of TB cases detected by about 5%, at a cost of $435 per new treatment, and reduced deaths by almost 50%, at a cost of $2284 per death; these represent an excellent return on investment based on prior studies. The authors highlight several key implications of their analysis. First, their estimates of the unit costs and overall cost-effectiveness of the active case-finding program appeared to be highly sensitive to the available public health capacity and efficiency of implementation in a particular setting (“supply” factors), as well as to the volume of undiagnosed and non-care-seeking TB patients (“demand” factors). Second, similar to previous studies, they found that the benefits of active case finding are likely to accrue over a more extended time period than is commonly realized – at least five years. Both these considerations are likely to be important as countries consider where and how they will or will not adopt such programs and evaluate initial implementation.

Thank you for the positive comments. 

Major Comments:

The analyses are well-presented and the figures and tables are very accessible to the reader. I have a few clarifying questions, as well as some questions about how these findings are packaged and presented to policymakers, as this is presumably one of the expected outcomes of publication.

Could the authors define what they mean by “generalized Zambian settings,” as compared with the settings of the CIDRZ ACF project? Are there contextual differences between them, or does this simply refer to nationwide scale-up?

Thank you for this question. We have clarified this on lines 87 and 150-151 of our revised manuscript: 

“a setting with TB epidemiology and economic conditions consistent with national averages” 

The manuscript presents the differences in unit costs per process measure between different time periods, but how variable were the costs per outcome (i.e. per TB diagnosis, per TB death) by quarter?

Thank you for this question. Using quarterly program yield data and expenditure/cost data, we calculated cost per programmatic yield (cost per TB diagnosis) for each quarter and presented these in S2 Table (also shown below). Cost per TB case detected/diagnosed and initiated on TB treatment varied from $312 to $658. While we did not estimate cost per death averted for each quarter, we performed probabilistic sensitivity analysis to ascertain uncertainty ranges (between $1,497 and $3,298 per TB death averted) of our primary incremental cost-effectiveness ratio (Table 3)

Types of services Average

 unit cost Cost per activity

 Jul-Sep, 2017 Oct-Dec, 2017 Jan-Mar 2018 Apr-Jun, 2018 Jul-Sep, 2018 Oct-Dec, 2018

Diagnosed as TB cases by X-ray $553 $409 $409 $575 $1,004 $461 $314

Diagnosed as TB cases by Xpert $755 $509 $509 $648 $980 $651 $985

Treatment initiated $435 $329 $329 $340 $533 $312 $658

Which of the factors influencing the temporal variations in cost are likely to be most easily modifiable? This information is contained in Figure 3, but it might be worth expanding on since HIV/TB mortality and symptom-associated passive care seeking may not be modifiable in ways that make ACF more cost-effective.

Thank you for these important comments. We have included a discussion on this matter in line 269-273:

“Since HIV/TB mortality and symptom-associated passive care seeking may not be easily modifiable, efforts to increase ACF service utilization in high prevalence areas through demand creation among key populations (e.g. people living with HIV or household contacts of people diagnosed with active TB) and/or optimization of intervention duration may be key strategies for program managers to improve the cost-effectiveness of ACF programs.”

While interesting, I wonder if the insights into the important effects of care-seeking behaviors on ACF cost-effectiveness tell the whole story. For example, the model implies that active case finding will be most cost-effective in settings where passive care-seeking behaviors and/or the efficiency of facility-based care is lowest. I imagine that this may commonly occur in facilities where the quality of care and service are poor, and often the low quality of care may also be well-known in the community. Yet, the treatment outcomes of patients identified during active case-finding may also depend on the quality of care in these facilities. If the goal is high rates of treatment coverage and cure, this kind of analysis may not capture these complexities. The authors hint at this in their call for more empiric data across the diagnosis and treatment cascade but I wonder if this specific aspect might be highlighted since the influence of care-seeking behaviors is such a salient finding of this analysis.

Thank you for the valuable comments. We revised the text in our discussion section to reflect the reviewer comments as the following (lines 279 – 283): 

“Moreover, the extent of care seeking and the value of ACF interventions can also be influenced by other health system factors (e.g. quality of care) and other patient factors (e.g. community awareness of TB and available services). Given the importance of these factors in determining the value of ACF interventions, we encourage future observational and modeling studies to investigate how and to what extent different health system and patient factors might impact the incremental benefits and costs of ACF.” 

In the last paragraph of the Results (Lines 189-198) and in the Discussion (Lines 227-230), the analysis uses temporal differences in cost and cost-effectiveness to generalize about such differences between settings. This seems potentially problematic since the word “settings” implies differences defined on factors other than time (e.g. geography). While time-based comparisons yields insights about how care can be delivered more efficiently to a fixed population (assumed stable, at least over the short term), comparisons between geographic settings should arguably account for selection differences between populations and for underlying heterogeneity in risks or behaviors within those populations, especially to the extent that the boundaries used to define populations geographically are arbitrary yet influence estimates of cost-effectiveness (i.e. different boundaries lead to different conclusions).

Thank you for the valuable comments. In response to this comment, we have extensively revised the last paragraph of the Results to discuss differences in “settings” as reflective of our sensitivity analyses (e.g. differentiated by patient care-seeking patterns, costs of ACF, and HIV/TB mortality, rather than relying on temporal differences to distinguish settings (lines 200-205):

“In settings where patients are more likely to passively seek care upon symptom onset (e.g., increase in the probability of symptom-associated care-seeking), the same ACF intervention would avert less than half of TB deaths (reduced from 750 to 300 per 100,000 population) and would require a longer time-frame to observe the full health benefit (16 vs. 28 months) as compared to the settings with lower care-seeking probabilities (S7 Figure). Depending on the costs of ACF and HIV/TB mortality levels, ICERs may vary as much as three times compared to the lowest ICER estimate (Figure 3).”

Similarly, in the last paragraph of the Discussion, we differentiate discussion on settings (as being distinguished by HIV/TB mortality, care seeking patterns, and operational context) and temporal dynamics (as might be considered by individual ACF programs), lines 305-311:

“The value of ACF may be optimized in settings where HIV/TB mortality is high, existing care seeking is infrequent, and when ACF interventions can cost-optimize patient screening by achieving operational efficiency. As these conditions may dynamically change throughout program implementation, individual ACF programs should carefully assess and actively monitor these indicators to identify the optimal timing and duration of operations.”

While the definition of boundaries may change the estimates of key epidemic or care seeking parameters (as it may change the patients’ profile), it will not change our conclusion (incremental cost effectiveness ratio of ACF vs PCF in which ACT is more cost effective compared to PCF and the cost effectiveness may differ by setting specific conditions). Nonetheless, we agree with the reviewer’s point and added the following text in the discussion section (line 257-273):

“Our study findings illustrate two distinctive implications for the dynamics of cost-effectiveness of ACF over time and place. First, S7 Figure demonstrates how the cost-effectiveness may change over time among the same (fixed) population in a given catchment area, driven by their symptom progression and associated care-seeking. This suggests that cost-effectiveness may differ by stage of implementation. For example, in the initial stage of program implementation, HIV/TB prevalence, care-seeking patterns, and associated symptom progression may be key drivers of cost-effectiveness, while operational efficiency may have greater influence on cost-effectiveness at later stages (as prevalence decreases and service volume increases over time). Our study estimated that incremental cost-effectiveness might fall about 30% (i.e. improved cost-effectiveness) from $3,200 in the first year to $2,284 in the fifth year (Figure 4). On the other hand, Figure 3 demonstrates to what extent the cost-effectiveness of ACF may differ across different regions and populations, driven by underlying heterogeneity in risks or behaviors within those populations. For example, another study that evaluated ACF in Cambodia estimated cost-effectiveness at $5,300 per death averted [6], more than two times greater than estimated here ($2,284 per death averted). Since HIV/TB mortality and symptom-associated passive care seeking may not be easily modifiable, efforts to increase ACF service utilization in high prevalence areas through demand creation among key populations (e.g. people living with HIV or household contacts of people diagnosed with active TB) and/or optimization of intervention duration may be key strategies for program managers to improve the cost-effectiveness of ACF programs.” 

Minor Comments:

The manuscript would benefit from a close reading for grammar and usage to improve its readability and clarity.

Thank you for this suggestion. We have carefully screened our manuscript for typos and corrected them accordingly.

Line 104: It is implied but not directly stated that specimens were collected for abnormal symptoms or abnormal mCXR; a little more detail on the ACF algorithm at this point in the Methods would be helpful for clarification.

Thank you for picking up on this issue. We have revised the wording as the following (lines 106 - 108):

“Both community-based outreach and facility-based team collected sputum samples on-spot from the patients showing abnormal symptoms or Chest Xray results for Xpert testing in the GPHC laboratory once patient was identified as a presumptive TB patient after initial symptom and mCXR screening.”

Table 1: The proportion of TB occurring in HIV patients is estimated at 59% (range 46-70%) is justified by an unpublished manuscript (Reference 21). While the Introduction cites similar numbers from the WHO 2020 TB control report, and certainly this estimate reflects case notification data, recent prevalence studies seem to suggest that the contribution of HIV-patients to TB may be lower in active case finding settings, perhaps because of the success of HIV programs in increasing HIV status awareness, and linkage to care which includes TB screening. See for example N. Kapata et al. PLOS ONE (2016).

Thank you for this suggestion. We replaced the original unpublished reference with the suggested references. 

References

1. Jo Y, Mirzoeva F, Chry M, Qin ZZ, Codlin A, Bobokhojaev O, et al. Standardized framework for evaluating costs of active case-finding programs: An analysis of two programs in Cambodia and Tajikistan. PLoS One. 2020;15(1):e0228216. doi: 10.1371/journal.pone.0228216. PubMed PMID: 31986183; PubMed Central PMCID: PMCPMC6984737.

2. Pooran A, Theron G, Zijenah L, Chanda D, Clowes P, Mwenge L, et al. Point of care Xpert MTB/RIF versus smear microscopy for tuberculosis diagnosis in southern African primary care clinics: a multicentre economic evaluation. Lancet Glob Health. 2019;7(6):e798-e807. doi: 10.1016/S2214-109X(19)30164-0. PubMed PMID: 31097281; PubMed Central PMCID: PMCPMC7197817.

3. Laurence YV, Griffiths UK, Vassall A. Costs to Health Services and the Patient of Treating Tuberculosis: A Systematic Literature Review. Pharmacoeconomics. 2015;33(9):939-55. doi: 10.1007/s40273-015-0279-6. PubMed PMID: 25939501; PubMed Central PMCID: PMCPMC4559093.

4. Woods B, Revill P, Sculpher M, Claxton K. Country-Level Cost-Effectiveness Thresholds: Initial Estimates and the Need for Further Research. Value Health. 2016;19(8):929-35. doi: 10.1016/j.jval.2016.02.017. PubMed PMID: 27987642; PubMed Central PMCID: PMCPMC5193154.

5. Kagujje M, Chilukutu L, Somwe P, Mutale J, Chiyenu K, Lumpa M, et al. Active TB case finding in a high burden setting; comparison of community and facility-based strategies in Lusaka, Zambia. PLoS One. 2020;15(9):e0237931. doi: 10.1371/journal.pone.0237931. PubMed PMID: 32911494; PubMed Central PMCID: PMCPMC7482928.

6. Dobler CC. Screening strategies for active tuberculosis: focus on cost-effectiveness. Clinicoecon Outcomes Res. 2016;8:335-47. doi: 10.2147/CEOR.S92244. PubMed PMID: 27418848; PubMed Central PMCID: PMCPMC4934456.

---

## [Decision Letter · Decision Letter 1]

10 Aug 2021

Costs and Cost-Effectiveness of a Comprehensive Tuberculosis Case Finding Strategy in Zambia

PONE-D-21-10534R1

Dear Dr. Sohn,

We’re pleased to inform you that your manuscript has been judged scientifically suitable for publication and will be formally accepted for publication once it meets all outstanding technical requirements.

Kind regards,

Kevin Schwartzman

Academic Editor

PLOS ONE

Additional Editor Comments (optional):

Thank you for carefully revising your manuscript in response to the reviewers' comments.

Reviewers' comments:

Reviewer's Responses to Questions

**Comments to the Author**

1. If the authors have adequately addressed your comments raised in a previous round of review and you feel that this manuscript is now acceptable for publication, you may indicate that here to bypass the “Comments to the Author” section, enter your conflict of interest statement in the “Confidential to Editor” section, and submit your "Accept" recommendation.

Reviewer #1: All comments have been addressed

Reviewer #2: All comments have been addressed

2. Is the manuscript technically sound, and do the data support the conclusions?

Reviewer #1: Yes

Reviewer #2: Yes

3. Has the statistical analysis been performed appropriately and rigorously? 

Reviewer #1: Yes

Reviewer #2: Yes

4. Have the authors made all data underlying the findings in their manuscript fully available?

Reviewer #1: Yes

Reviewer #2: Yes

5. Is the manuscript presented in an intelligible fashion and written in standard English?

Reviewer #1: Yes

Reviewer #2: Yes

6. Review Comments to the Author

Reviewer #1: All of my comments have been well addressed, and I have no further comments. Congratulations on this excellent paper!

Reviewer #2: The authors have responded thoughtfully and comprehensively to the questions and concerns that I raised in the initial review.

7. PLOS authors have the option to publish the peer review history of their article (what does this mean?). If published, this will include your full peer review and any attached files.

Reviewer #1: **Yes: **Sedona Sweeney

Reviewer #2: No

---

## [Editor Report · Acceptance letter]

25 Aug 2021

PONE-D-21-10534R1 

Costs and Cost-Effectiveness of a Comprehensive Tuberculosis Case Finding Strategy in Zambia 

Dear Dr. Sohn:

I'm pleased to inform you that your manuscript has been deemed suitable for publication in PLOS ONE. Congratulations! Your manuscript is now with our production department. 

Kind regards, 

on behalf of

Dr. Kevin Schwartzman 

Academic Editor

PLOS ONE